# Effect of Dietary Orange Peel Meal and Multi-Enzymes on Productive, Physiological and Nutritional Responses of Broiler Chickens

**DOI:** 10.3390/ani13152473

**Published:** 2023-07-31

**Authors:** Maha A. Abd El Latif, Ahmed A. A. Abdel-Wareth, Milton Daley, Jayant Lohakare

**Affiliations:** 1Department of Animal and Poultry Production, Faculty of Agriculture, Minia University, Minia 61519, Egypt; maha.omr@mu.edu.eg; 2Department of Animal and Poultry Production, Faculty of Agriculture, South Valley University, Qena 83523, Egypt; 3Poultry Center, Cooperative Agricultural Research Center, Prairie View A & M University, Prairie View, TX 77446, USA; mdaley@pvamu.edu

**Keywords:** broiler chicken, feed additives, performance, sustainability, blood biochemistry, antioxidants

## Abstract

**Simple Summary:**

Agricultural byproducts can play a significant role in poultry nutrition as an alternative to traditional ingredient resources. Orange peel meal has the potential to be a source of essential nutrients and reduce oxidative stress in chickens. The addition of dietary orange peel meal to broiler chicken diets at the four graded concentrations of 0, 80, 160, and 240 g kg^‒1^ with or without multi-enzymes was evaluated in this study. The study concluded that orange peel meal improved body weight gain, the feed conversion ratio (FCR) during the grower phase, antioxidant capacity, and decreased abdominal fat; multi-enzyme addition improved body weight gain, FCR during the grower phase, crude fiber digestibility, T3 levels, and decreased the fat percentage; and an interaction effect was observed only for SOD levels, where SOD levels decreased due to the addition of enzymes.

**Abstract:**

The objective of the study was to evaluate the impact of various concentrations of orange (*Citrus sinensis*) peel meal (OPM), with or without the supplementation of multi-enzymes, on the growth performance, nutrient digestibility, antioxidant properties, and blood metabolic profile of broiler chickens. The experiment was conducted on 240 one-day-old Arbor Acres broiler chicks, assigned to eight dietary treatments with 30 broilers per treatment group. Four dietary orange peel meal (OPM) concentrations were supplemented, namely, the control (without OPM), and with 80, 160, and 240 g/kg of the diet. To each of these diets was added two concentrations of multi-enzyme inclusion (0 or 0.6 g as a combination of 0.5 g of Nutrikem and 0.1 g Optiphos per kg diet) in a completely randomized design in a 4 × 2 factorial arrangement. The experiment lasted until 42 days of age. Body weight gain (BWG) was influenced during the grower period (22–42 days) and the overall period (0–42 days), and the feed conversion ratio (FCR) was significantly improved by supplementations of OPM compared with the control for 22–42 days and overall (0–42 days) periods. Moreover, BWG, FCR during the grower and overall periods, and crude fiber digestibility were improved (*p* < 0.01) by multi-enzyme supplementation compared to the non-supplemented groups. Broilers with diets supplemented with OPM had considerably lower abdominal fat (*p* < 0.01) than the control. In addition, when compared to the non-supplemented enzyme group, serum T3 and T3/T4 ratios were significantly improved in response to enzyme addition. When compared to the control group, superoxide dismutase (SOD) was significantly higher in the OPM groups, showing the largest improvement in antioxidant response. Interaction effects were observed only for serum SOD levels. Based on our findings, it is recommended that OPM be used as a feed supplement for raising broilers, and adding 0.6 m g/kg of multi-enzymes could provide additional benefits to the performance of broilers.

## 1. Introduction 

Supplemental nutrients that are beneficial to poultry growth and health may be present in plant waste products utilized in the human food business. Chemistry breakthroughs have led to opportunities to explore plant medicines, particularly the discovery of phytochemical components that can improve health and performance. Animal feeding using agricultural byproducts is a major source of increasing profitability. For poultry, orange peel meal (OPM) is a potential source of important nutrients and natural antioxidants. Antioxidants are found in abundance in fruits and vegetables, and they can neutralize free radicals and convert them to harmless molecules [1]. According to Byer et al. [2], increasing antioxidant levels lowers free radical reactions, which may benefit cell activity. OPM has antioxidant qualities, according to Mona and Hanan [3], which can increase the productivity of broiler chickens. 

Nutritional studies on monogastric animals have thus far demonstrated that a meal of sun-dried sweet orange peels of *Citrus sinensis* can substitute up to 20% of dietary corn in broiler diets without having side effects on their performance [4]. Enzyme supplements have been used to improve the growth performance of broilers [5]. The digestibility of the feed is enhanced and the viscosity of the digesta is reduced when exogenous enzymes are added to the diet in the grower phase of broiler chickens [5,6]. Furthermore, enzyme supplementation may enhance growth performance by lowering intestinal viscosity and modulating gut microbiota [7,8]. Nutritionally, economically, and environmentally, supplementing poultry diets with multi-enzymes is justified. The nutritional value of diets will be improved by the strategic development of appropriate enzyme combinations based on the diet’s composition. Optiphos is a 6-phytase enzyme product produced from an *Escherichia coli* gene expressed in Pichia pastoris. It is provided in a coated form for enhanced recovery, post-steam conditioning, and pelleting. Lysophospholipids are used as a matrix to encase the NSP enzymes and multi-protease that make up the NUTRIKEM XL Pro enzyme. These sustain the productive performance of the birds and aid to increase nutrient absorption from raw materials that are poorly digested, while also giving broiler producers a clear economic benefit. However, there is limited information on the proper enzyme or enzyme combination for broilers based on corn–soybean diets with soybean meal partially supplemented with OPM. This study was conducted to study the effect of different OPM concentrations, with or without enzyme supplementation, on the growth performance, nutrient digestibility, antioxidant properties, and blood metabolic profile of broiler chickens.

## 2. Material and Methods 

### 2.1. Animals, Diets, and Experimental Design

The experiment was conducted on 240 straight-run one-day-old Arbor Acres broiler chicks, assigned to eight dietary treatments. Eight treatment diets were used: four diets with 4 levels of OPM with and without multi-enzyme supplementation. Four levels of OPM at 0, 80, 160, and 240 g/kg of the diet, and other four diets with multi-enzymes (0.6 g) in addition to the above four levels of OPM, were used in a randomized complete block design in a 4 × 2 factorial arrangement. The 0.6 g multi-enzymes used was a combination of 0.5 g of Nutrikem and 0.1 g Optiphos per kg diet. Each group had 6 replicates (5 birds/pen). The experiment lasted until 42 days of age. Birds were fed mash diets to meet the nutrient requirements according to Arbor Acres broiler chicken recommendations (Table 1) during the starter (1–21 d) and grower (22–42 d) periods, respectively. Each 1 g Optiphos contains Phytase enzyme (2500 OUT) and calcium carbonate up to 1 g (Huvepharma^®^ Company, Sofia, Bulgaria). NUTRIKEM XL Pro is a comprehensive enzyme solution containing multiple NSPases, patented Multi protease, and Lysophospholipids (Kemin Industries, Inc., Valley Center, CA, USA).

### 2.2. Experimental Conditions 

During the experiment, the birds were housed and handled according to the Minia University, Institutional Animal Care and Use Committee’s recommendations (AGRMU-0040123). The birds were kept in deep litter systems with wood shavings as litter material in pens with dimensions of 130, 80, and 70 cm in length, width, and height, respectively. Water and feed were freely available to the birds using waterers and manual feeders, respectively. The lighting schedule and climatic conditions were carried out according to commercial recommendations. All chicks were kept under the same management guidelines and the temperature of 34 °C was set for the first week, gradually dropping to 24 °C by the fourth week and afterward. 

### 2.3. The Orange Peel Meal (OPM) Preparation 

Fresh oranges (*Citrus sinensis)* were acquired from local vendors, peeled and cut with a knife, and then laid out to dry until crispy. The peels were ground into powder after drying. Moisture, ash, crude protein (CP), crude fiber (CF), ether extract (EE), and nitrogen-free extract (NFE) were determined in a sample of OPM analyzed in the Animal Nutrition Laboratory of Minia University’s Faculty of Agriculture according to AOAC [9].

Supplemental multi-enzymes, Nutrikem and Optiphos, were purchased from United Medvit company to be used as feed additives. Nutrikem is a mixture containing Xylanase: 10,000 U/g; β glucanase: 1175 U/g; α-amylase: 200 U/g; cellulase: 2000 U/g; protease: 225 U/g; carrier; lecithin: 12.5%; silicic acid: 7.5%; bentonite: 2.5%; vegetable oil: 0.5%; and calcium carbonate up to 100%. Optiphos 5000 composition: each 1 g contains: 6-phytase (Pichia pastoris) 23%, pregelatinized starch 1%, and wheat flour up to 100%. 

### 2.4. Performance Indices

To evaluate body weight gain (BWG), the birds were weighed at the start of the experiment, and 21 and 42 days later, and feed intake was estimated as the difference between the offered and residual feed for each replicate at (0–21), (22–42), and (0–42) days of age. Based on BWG and FI data, the feed conversion ratio (FCR) was calculated, and the mortalities were also measured. 

### 2.5. Nutrient Digestibility 

Metabolic cages were used to house six birds from each treatment (one per replicate that represented the pen in each metabolic cage); each cage had an automatic drinking nipple and a manual feeder, and its width, length, and height measured 70 cm, 60 cm, and 40 cm, respectively. After the adaptation period from 35 to 38 days old, the measuring period lasted another four days for total collection of the excreta. Throughout the entire time, the amount of feed consumed was carefully recorded, and excreta were quantitatively collected at about 08:00 a.m. before the next feeding time. All excreta that were collected during the digestibility trial were mixed, dried at 60 °C for 48 h before analyses, and then representative excreta samples were ground for chemical analyses. The feed and excreta were analyzed for moisture by oven drying (method No. 930.15), ash by incineration (method No. 942.05), protein by Kjeldahl (method No. 984.13), ether extract by Soxhlet fat analysis (method No. 954.02), and crude fiber (method No. 978.10) method described by the AOAC International [9]. The following formula was used to determine nutrient digestibility:$$\mathrm{Apparent}\;\mathrm{digestibility}\;\left(\%\right)=\frac{\left(\mathrm{Nutrient}\;\mathrm{ingested}-\mathrm{Nutrient}\;\mathrm{excreted}\;\mathrm{in}\;\mathrm{feces}\right)}{\mathrm{Nutrient}\;\mathrm{ingested}\;}\times 100$$

### 2.6. Carcass Criteria 

After 42 days, 24 birds were randomly selected from the group (4 birds per replicate) and were processed to evaluate the internal organs and carcass standards. Individually weighed birds were humanely sacrificed, allowed to bleed, and then harvested. The rest of the body was weighed after the neck, head, viscera, shanks, spleen, digestive tract, heart, gizzard, and belly fat were removed.

The dressing percentage was determined by dividing the carcass and giblet weight by the live weight. Each bird’s heart, spleen, empty gizzard, and abdominal fat were weighed individually and represented as a percentage of live body weight.

### 2.7. Serobiochemical Assays

At the end of the experiment, the serum was obtained by drawing blood from the wing veins of 24 birds per treatment using sterile needles and syringes (42 days of age). The blood was centrifuged at room temperature (3000× *g*) for 15 min. Until it was analyzed, the serum was collected in tubes and stored at −20 °C. The T3 and T4 hormones were determined by immunoassay techniques using RIA kits at Atomic Energy Authority laboratory in Egypt. Antioxidants such as Malondialdehyde (MDA), Superoxide dismutase (SOD), Glutathione (GSH), Ascorbic acid (Vit. C), and Tocopherol (Vit. E) were determined using colorimetry assays, [11,12,13,14], respectively. Serum biochemical parameters: total protein, albumin, glucose, total bilirubin, alanine aminotransferase (ALT), aspartate aminotransferase (AST), triglycerides (TG), total cholesterol (TCho), high-density lipoprotein cholesterol (HDL-cholesterol), low-density lipoprotein cholesterol (LDL-cholesterol) were analyzed. These parameters were analyzed using commercial kits, (Bio-diagnostic, Cairo, Egypt). Globulin (Glob), ALT/AST ratio, T3/T4 ratio, and very low-density lipoprotein (VLDL) were calculated. Amylase activity using the method of Somogyi [14] and protease activity were analyzed using the method of Lynn and Clevette-Radford [15].

### 2.8. Statistical Analysis 

The main effects (OPM, multi-enzymes) and interaction effects between treatments were tested. Pen was the experimental unit for all the measured parameters. All data were evaluated for normal distribution (W > 0.05) using the Shapiro–Wilk test. Then, two-way ANOVA was performed using SAS 9.2 [16]. Values were expressed as the mean and standard error of the mean (SEM). Treatment means were compared using Duncan’s Multiple Range Test [17], and differences were considered significant at *p* < 0.05. 

## 3. Results 

### 3.1. Growth Performance

The effect of feeding OPM with or without multi-enzymes on growth performance in broiler chickens is presented in Table 2. Body weight gain was increased (*p* < 0.01) by supplementations of OPM or multi-enzymes alone to broiler diets during the period of 22 to 42 days and 0 to 42 days of age. Likewise, the feed conversion ratio (FCR) was improved (*p* < 0.05) by supplementations of OPM or multi-enzymes alone compared with the control during the period of 22–42 days and 0 to 42 days of age. The best BWG and FCR were observed in 160 g/kg OPM for the period of 22 to 42 days and 0 to 42 days compared to others, but there were no differences between the 80 and 240 g/kg OPM groups. However, supplementation of OPM did not impact the feed intake during the experimental periods. The interaction effects of OPM and multi-enzymes did not show any significant response on growth performance in broiler chicks (Table 2).

### 3.2. Nutrient Digestibility 

The effect of feeding OPM with or without multi-enzymes on nutrient digestibility in broiler chicks is presented in Table 3. Ether extract digestibility was improved (*p* < 0.01) by supplementations of OPM at 80 and 160 g/kg compared to other treatments (control and 240 g/kg). Moreover, crude fiber digestibility was improved (*p* < 0.01) by multi-enzyme supplementation compared to non-enzyme-supplemented groups. However, there were no differences in dry matter, organic matter, or crude protein digestibility by supplementation compared to the control. The interaction effects of OPM and multi-enzymes did not show any significant responses on nutrient digestibility in broiler chicks.

### 3.3. Carcass Criteria 

The effect of feeding OPM with or without multi-enzymes on carcass criteria in broiler chicks is presented in Table 4. The 160 and 240 g/kg OPM improved the dressing percentage compared to the control; however, the dressing percentage was not different in birds fed the 80 g/kg OPM diet according to data in Table 4. Moreover, the abdominal fat percentage was decreased (*p* < 0.05) by OPM supplementation as well as by multi-enzyme supplementation compared with their non-supplemented counterparts. Additionally, heart percentage was decreased (*p* < 0.05) by multi-enzyme supplementation compared with the non-enzyme group. However, there were no differences in internal organs such as liver, heart, gizzard, and giblet percentages by supplementations compared to the control. The interaction effects of OPM and multi-enzymes did not show any significant responses on carcass criteria in broiler chicks.

### 3.4. Serum Biochemistry 

Table 5, Table 6 and Table 7 show the effect of feeding OPM with or without multi-enzymes on serum biochemistry in broiler chicks. In comparison to the 0 g/kg OPM and 240 g/kg OPM treatments, the glucose levels were low in the 160 g/kg OPM treatment. However, the glucose levels in the birds fed the 0 OPM and 240 g/kg OPM did not differ. Bilirubin levels were significantly decreased in the 80 g/kg OPM-supplemented group compared to the 160 and 240 g/kg OPM groups (Table 5). Supplementations had no significant effect on total protein, albumin, globulin, or the albumin:globulin ratio when compared to the control. Broiler diets supplemented with OPM with or without multi-enzymes had no effect on serum concentrations of ALT and AST activity compared to the control; however, there was a tendency effect (*p* < 0.10 > 0.05) on serum AST concentrations due to OPM, multi-enzymes, or their interaction (Table 6). Furthermore, amylase and protease activity were not influenced in response to OPM and multi-enzyme supplementation as compared to the control group (Table 6). In addition, serum T3 levels were considerably improved (*p* < 0.05) in response to OPM, and with multi-enzymes compared to their counterparts (Table 6). The OPM levels in broiler chicks had a significant impact on serum total cholesterol, triglycerides, HDL, and VLDL (Table 7). However, LDL did not significantly change between the treatments. 

### 3.5. Antioxidant Profile

The effect of feeding OPM with or without multi-enzymes on the antioxidant profile in broiler chicks is shown in Table 8. Broilers fed 240 g/kg OPM had higher concentrations of Vit. C, and superoxide dismutase compared to the control. However, superoxide dismutase exhibits interaction effects between the groups. Superoxide dismutase was lower (*p* < 0.05) in the multi-enzyme supplemented groups than their non-supplemented counterparts for the 80, 160, and 240 g/kg groups, indicating the effects of multi-enzymes in antioxidant response.

## 4. Discussion

There are presently few reports on commercial diet dilution using low-cost fibrous materials such as orange peel meal and feed additives such as multi-enzymes for broilers. Citrus peels are higher in nutritional quality than other low-cost fibrous agri-byproducts because they contain bioactive chemicals such as antioxidants, polyphenols, carotenes, and flavonoids [1,2]. Antibacterial activity against pathogenic bacteria is known for these phytogenic compounds [4]. In this study, we found that supplementing OPM enhanced BWG during the grower and overall periods and FCR considerably when compared to the control. The FCR, as is widely known, is directly proportional to daily body weight gain and total feed consumption. The improvement in body weight gain by OPM could be related to a decrease in pathogens in the gastrointestinal tract, which allows nutrients to be absorbed by the birds [18]. Vlaicu et al. [19] showed that broiler chickens fed a diet supplemented with 2% OPM had the highest body weight and the lowest trend of FCR 

Furthermore, Mourao et al. [20] found that including citrus pulp into a chicken diet improved feed the conversion ratio when compared to the control, similar to our results. Dry orange peel powder diets had slightly better growth parameters, and adding multi-enzymes to the diet seems to improve broiler BWG without affecting feed efficiency [21]. The effect of orange peel may be attributed to its antioxidant activity. Since a common basal diet was not used, different feed ingredients were used to make the treatment diets. The variability of ingredients may also have an impact on the obtained results. The addition of an enzyme supplement to a broiler diet containing 4% olive meal resulted in an optimal combination for supporting broiler growth performance and carcass characteristics without negatively influencing blood biochemistry [22]. When compared to the control diet, the addition of enzymes boosted body weight gain and improved the feed conversion ratio [23]. Because such meals contain a high percentage of non-starch polysaccharides, the enzymes help monogastric animals perform better [24]. The digestibility of the digesta is increased by the inclusion of exogenous enzymes in the grower phase [5]. Furthermore, enzyme supplementation may enhance growth performance by lowering intestinal viscosity and modulating gut flora [7,8]. In addition, the supplementation of OPM at 80 and 160 g/kg enhanced ether extract digestibility significantly when compared to other treatments in the current investigation. When compared to the enzyme non-supplemented group, enzyme treatment improved crude fiber digestibility substantially. OPM and enzyme interactions, however, had no appreciable impact on the nutrient digestibility of broiler chicks. To improve fiber digestion or solubilize phytic phosphorus, exogenous enzymes are commonly added to broiler diets, or feed byproducts, reducing the negative impact of these substances on broiler performance [25,26]. Furthermore, dietary enzyme supplementation has been shown to increase nutritional value by disrupting non-starch polysaccharides, improving nutrient digestion, and reducing microbial fermentation in the small intestine [27]. The amount of nutrients obtained by the bird digestive tract from these feed byproducts is considerably increased when enzymes are added [28,29]. 

In the current study, the dressing percentage increased and the abdominal fat percentage decreased through supplementations of OPM; and multi-enzymes decreased the fat percentage compared with their counterparts. The improvements in growth performances may be primarily responsible for the increase in carcass yield in the OPM-supplemented groups. The BWG and the feed conversion ratio were marginally better with dry orange peel powder diets, and adding multi-enzymes to the diet appeared to increase the broiler carcass yield and decrease abdominal fat [21]. Orange peel’s impact may be linked to its antioxidant activity, which may reduce abdominal fat. A broiler diet comprising 4% olive meal and an enzyme supplement produced the best results for the characteristics of the broiler carcass [22]. The inclusion of enzymes in broiler diets enhanced carcass weight and decreased abdominal fat when compared to the control diet [23]. The enzymes aid birds in functioning better since such diets contain a large proportion of non-starch polysaccharides [24]. The results partially corroborated those of Ebrahimi et al. [30], who reported that the length of the jejunum and ileum were significantly increased by dried *Citrus sinensis* peel; however, the abdominal fat percentage was significantly reduced when compared to the control group. 

In the current study, OPM supplementation at 240 g/kg significantly increased serum glucose and total bilirubin compared to others, but this was no different to the control group. Multi-enzyme supplementation also increased serum glucose concentrations compared to the non-supplemented group. Broiler diets supplemented with OPM with or without multi-enzymes had no effects on serum concentrations of ALT and AST activity compared to the control. In addition, when compared to the control group, serum T3 was considerably improved in response to OPM supplementation and it was higher in multi-enzyme treatments compared to non-enzyme treatments. The possible use of OPM in broiler feeding and its effects on the serum biochemistry of broilers are poorly covered in the literature. The concentrations of ALT and AST were found to be within normal limits [31,32,33,34], showing that the liver and kidneys were operating normally in studies where dried orange or citrus pulp was used to feed chickens. In contrast to our results, citrus pulp supplementation at the level of 6% in the broiler ration did not show any negative effect on blood glucose levels [35]. The blood parameters were all within the normal range [36]. It was also noted that adding orange peel extract and lemon peel extract had no effects on serum total protein, albumin, or globulin levels [37], similar to our results. Total protein, albumin, and globulin levels were unaffected by the addition of citrus waste to the diet in line with our results, while plasma AST levels dropped linearly as the amount of citrus waste in the diet increased [38], contrary to our results. In the current study, the OPM and enzyme interaction effects tended to be significant for serum total cholesterol, triglycerides, low-density lipoprotein, and VLDL. Citrus waste supplementation in the broiler diet reduced blood cholesterol and triglyceride levels [33]. Broilers fed a diet containing 2% olive meal without the supplemental enzyme had higher total cholesterol than those fed a diet containing 4% olive meal with the enzyme; however, blood LDL and HDL cholesterol, triglycerides, total protein, albumin, and glucose were not different between treatment groups [22]. Supplementation of enzymes significantly decreased ALT, AST, and MDA, compared to the control group [23]. Broilers fed OPM concentrations in the current study had higher levels of vitamin C and superoxide dismutase compared to the control, indicating the largest improvement in antioxidant response. In addition, in response to OPM with or without multi-enzyme supplementation, amylase and protease activity were not different compared to the control group, in the present study. 

Alzawqari et al. [38] indicated that dried sweet orange peel may positively modify serum T3, T4, AST, and ALT and the antioxidant status in broiler chickens. Citrus fruit is a significant source of pectin [39]. Our findings corroborated with Nobakht [40] that dried citrus pulp had favorable effects on the decrease in blood cholesterol, LDL, and VLDL. Furthermore, Abbasi et al. [32] reported reduced LDL, HDL, and triglycerides with no effect on blood glucose and cholesterol in broilers in response to dietary treatment with C. sinensis pulp, and they hypothesized that vitamin C and other substances found in the pulp of citrus fruits may be the reason for the altered blood metabolites. The literature provides scant information about the potential use of OPM in broiler feeding and its effects on the antioxidant traits of broilers, suggesting that additional research is required to determine the mode of action of OPM on antioxidant traits. Catalase and SOD activity increased non-significantly as the amount of citrus waste in broiler diets increased, while lipid peroxidation, glutathione peroxidase, and glutathione activities decreased up to 5% in citrus waste-fed groups [41]. Vitamin E and C activity, as well as glutathione activities and serum vitamin C levels, were found to be highest in birds fed citrus waste-based diets supplemented with enzymes [32]. Antioxidant levels in broiler blood, such as SOD, catalase, and glutathione peroxidase increased as the amount of citrus waste in the feed increased [42], in line with our results where SOD activities increased with OPM supplementation. Moreover, the antioxidant action of the OPM could be attributed to the flavones [43,44] present in them. One of the most significant flavanones to be extracted from orange peel, hesperidins, has demonstrated diuretic and antioxidant effects [45]. Orange peel’s antioxidant properties are linked to its free aromatic OH-groups, which can donate a H atom to help lower the number of free radicals [46]. 

## 5. Conclusions

Based on the observed results, dietary OPM of up to 260 g/kg may be beneficial to improve BWG and FCR for the grower and overall periods, and to improve ether extract digestibility, dressing percent, abdominal fat percent, serum glucose, and SOD levels while having no detrimental impacts on other blood biochemistry of broilers. Therefore, based on the results, it is recommended to use OPM as a feed supplement for raising broilers, and adding 0.6 m g/kg of multi-enzymes could provide additional benefits in the performance of broilers.

## Figures and Tables

**Table 1 animals-13-02473-t001:** Composition and analysis of the experimental diets.

Ingredients %	Starter Diets				Grower Diets			
	Control	OPM80	OPM160	OPM240	Control	OPM80	OPM160	OPM240
Yellow corn	52.20	47.00	42.00	34.00	58.50	51.00	45.00	37.00
Soybean meal, CP 44%	34.60	32.00	28.80	25.80	29.00	27.20	24.00	22.00
Corn gluten meal, CP 60%	5.70	7.00	8.00	10.00	4.00	5.00	6.70	8.00
Orange peel meal *	0.00	8.00	16.00	24.00	0.00	8.00	16.00	24.00
Wheat bran	0.00	0.00	0.00	2.00	0.00	1.00	1.30	2.30
Oil	3.50	3.00	2.50	2.00	4.50	4.20	4.00	4.20
Di-calcium phosphate limestone	2.50 1.00	1.70 0.80	1.50 0.70	1.2 0.50	2.40 0.90	2.20 0.90	1.80 0.70	1.50 0.50
Common salt	0.25	0.25	0.25	0.25	0.25	0.25	0.25	0.25
minerals &vitamins **	0.25	0.25	0.25	0.25	0.25	0.25	0.25	0.25
L-lysine	0.00	0.00	0.00	0.00	0.00	0.00	0.00	0.00
Total	100.00	100.00	100.00	100.00	100.00	100.00	100.00	100.00
Calculated analysis:								
ME (kcal/kg)	3063	3089	3072	3045	3151	3136	3166	3183
CP%	23.19	23.30	22.57	22.50	20.25	20.16	19.83	20.11
CF%	3.64	4.50	5.21	6.25	3.37	4.34	5.11	6.05
Ca%	1.05	1.14	1.24	1.39	0.96	1.28	1.25	1.36
Av. ph%	0.73	0.67	0.68	0.69	0.71	0.79	0.68	0.67
Lysine%	1.13	1.15	0.95	0.99	0.97	1.01	0.87	0.85
Methionine%	0.38	0.40	0.35	0.40	0.35	0.30	0.40	0.30
Cysteine%	0.22	0.30	0.21	0.33	0.24	0.25	0.32	0.20
Laboratory analysis:								
DM ^1^%	93.78	93.50	92.93	92.65	92.16	92.35	92.35	91.87
OM ^2^%	85.98	85.63	85.01	84.62	85.01	85.44	84.80	84.55
CP%	22.81	22.52	22.09	22.21	20.00	19.70	19.55	19.52
CF%	3.54	4.20	5.04	5.92	3.50	4.03	4.77	5.79
EE%	7.67	7.32	6.80	6.70	9.12	10.4	10.45	10.53
Ash%	7.80	7.87	7.92	7.64	7.15	6.91	7.51	7.32
NFE ^3^%	58.18	58.09	58.15	57.53	60.23	58.96	57.72	56.84

DM: dry matter, OM: organic matter, CP: crude protein, CF: crude fiber, EE: ether extract, NFE: * nitrogen-free extract; Laboratory analysis of orange peel meal (OPM) was performed according to AOAC [9]: DM 84.46%, OM 78.33%, Ash 6.12%, CP 6.45%, CF 13.18%, EE 6.21%, and NFE 68.04%. ME kcal/kg = 3157.08 was calculated according to Pauzenga [10]. ** Each 1 kg Premix contained: Vit A 3,350,000 IU, Vit D3 760,000 IU, Vit E 6700 IU, Vit K3 335 mg, Vit B1 334 mg, Vit B2 1670 mg, Vit B6 500 mg, Vit B12 3.4 mg, Niacin 10,000 mg, Ca Pantothenate 3334 mg, Biotin 16.7 mg, and Folic acid 334 mg; Trace minerals: Iron 13,350 mg, Copper 3335 mg, Zinc 16,700 mg, Manganese 25,000 mg, Iodine 500 mg, Cobalt 84 mg, and Selenium 100 mg; Additives: Ethoxyquine 600 mg, and Carrier (CaCO_3_) up to 1 kg; ^1^ DM = 100 − moisture%, ^2^ OM = DM% − ash%, ^3^ Nitrogen-free extract (NFE) = 100 − (CP% + CF% + EE% + Ash%).

**Table 2 animals-13-02473-t002:** Effect of dietary orange peel meal with or without multi-enzymes on growth performance of broiler chicks.

Factors	Body Weight Gain, g			Feed Intake, g			Feed Conversion Ratio		
	0–21 d	22–42 d	0–42 d	0–21 d	22–42 d	0–42 d	0–21 d	22–42 d	0–42 d
OPM, g/kg									
0	618	1487 ^c^	2105 ^c^	1097	2996	4093	1.78	2.02 ^a^	1.94 ^a^
80	664	1643 ^b^	2306 ^b^	1123	2867	3990	1.69	1.73 ^bc^	1.72 ^b^
160	640	1826 ^a^	2468 ^a^	1155	3024	4180	1.81	1.66 ^c^	1.70 ^b^
240	689	1704 ^b^	2394 ^b^	1136	3036	4172	1.66	1.78 ^b^	1.75 ^b^
Multi-enzymes, g/kg									
0	641	1620 ^b^	2262 ^b^	1132	2956	4089	1.77	1.84 ^a^	1.81 ^a^
0.6	665	1710 ^a^	2375 ^a^	1123	2991	4115	1.70	1.76 ^b^	1.74 ^b^
OPM + Multi-enzymes									
0 + 0	622	1450	2072	1123	3019	4142	1.82	2.08	2.00
0 + 0.6	614	1525	2139	1072	2973	4045	1.75	1.95	1.90
80 + 0	642	1565	2207	1103	2773	3876	1.72	1.75	1.74
80 + 0.6	684	1721	2406	1142	2998	4140	1.67	1.71	1.70
160 + 0	620	1821	2453	1159	3064	4222	1.87	1.67	1.72
160 + 0.6	660	1831	2481	1152	2985	4138	1.75	1.65	1.67
240 + 0	680	1635	2315	1145	3006	4222	1.69	1.84	1.79
240 + 0.6	700	1773	2472	1127	3066	4193	1.63	1.73	1.70
SEM	13.09	31.71	36.81	17.01	27.49	32.36	0.03	0.03	0.03
*p*-value									
OPM	0.323	˂0.001	˂0.001	0.751	0.063	0.058	0.146	˂0.001	˂0.001
Multi-enzymes	0.407	0.023	0.027	0.818	0.444	0.641	0.165	0.020	0.015
OPM × Multi-enzymes	0.912	0.364	0.549	0.874	0.143	0.137	0.962	0.557	0.808

^a,b,c,^ Treatment means in a column with different superscripts differ (*p* ˂ 0.05); OPM: dietary orange peel meal; OPM × Multi-enzyme interaction effect; SEM: Standard Error of Means. Each treatment had 6 replicate pens with 5 birds each. Pens were the experimental units for statistical analysis.

**Table 3 animals-13-02473-t003:** Effect of dietary orange peel meal with or without multi-enzymes on nutrient digestibility of broiler chicks.

Factors	DM%	OM%	CP%	CF%	EE%	NFE%
OPM, g/kg						
0	77.21	78.72	83.73	33.59	94.55 ^b^	78.79
80	82.92	82.05	85.59	42.42	96.06 ^a^	80.83
160	75.96	76.27	81.29	40.18	96.35 ^a^	75.67
240	78.38	80.67	84.58	37.94	94.11 ^b^	80.41
Multi-enzymes, g/kg						
0	79.00	79.12	82.69	33.31 ^b^	95.21	78.57
0.6	78.23	79.74	84.89	43.76 ^a^	95.32	79.29
OPM + Multi-enzymes						
0 + 0	77.71	79.41	83.37	31.16	94.12	79.35
0 + 0.6	76.71	78.04	84.08	36.01	94.98	78.24
80 + 0	81.65	81.18	83.60	34.77	96.59	79.38
80 + 0.6	84.19	82.92	87.57	50.07	95.53	82.27
160 + 0	74.03	73.06	79.31	34.56	93.26	72.74
160 + 0.6	77.90	79.48	83.26	45.80	95.43	78.60
240 + 0	76.68	82.82	84.44	32.74	96.85	82.79
240 + 0.6	80.07	78.52	84.70	43.14	95.35	78.03
SEM	3.013	2.482	1.629	3.725	0.564	2.350
*p*-value						
OPM	0.151	0.148	0.093	0.147	0.007	0.157
Multi-enzymes	0.724	0.727	0.074	0.001	0.770	0.670
OPM × Multi-enzymes	0.636	0.206	0.520	0.584	0.106	0.163

^a,b^ Treatment means in a column with different superscripts differ (*p* ˂ 0.05). OPM: dietary orange peel meal; OPM × Multi-enzyme interaction effect; SEM: Standard Error of Means; DM: dry matter; OM: organic matter; CP: crude protein; CF: crude fiber; EE: ether extract; NFE: nitrogen-free extract. Each treatment had 6 replicate pens with 1 bird each. Pens were the experimental units for statistical analysis.

**Table 4 animals-13-02473-t004:** Effect of dietary orange peel meal with or without multi-enzymes on carcass characteristics of broiler chicks.

Factors	Body Weight, g	Dressing%	Liver %	Gizzard %	Heart%	Fat %	Giblets %
OPM, g/kg							
0	2120	74.78 ^b^	3.08	2.38	0.63	1.91 ^a^	7.30
80	2155	76.68 ^ab^	3.26	2.06	0.66	1.43 ^b^	7.41
160	2302	79.77 ^a^	2.68	2.16	0.59	1.39 ^b^	7.11
240	2291	80.56 ^a^	2.60	2.03	0.59	1.34 ^b^	6.31
Multi-enzymes, g/kg							
0	2180	76.61	2.89	2.23	0.66 ^a^	1.59 ^a^	7.19
0.6	2254	79.28	2.92	2.09	0.53 ^b^	1.38 ^b^	6.88
OPM + Multi-enzymes							
0 + 0	2037	72.27	2.98	2.78	0.74	1.41	7.92
0 + 0.6	2203	77.31	3.18	1.98	0.51	1.19	6.68
80 + 0	2105	75.99	3.46	1.96	0.66	1.52	7.60
80 + 0.6	2206	77.37	3.06	2.17	0.66	1.33	7.21
160 + 0	2203	78.69	2.44	2.06	0.63	1.62	6.71
160 + 0.6	2400	80.84	2.91	2.27	0.56	1.76	7.51
240 + 0	2375	79.51	2.66	2.12	0.63	1.84	6.54
240 + 0.6	2406	81.60	2.54	1.94	0.38	1.24	6.09
SEM	165	2.45	0.287	0.207	0.067	0.131	0.457
*p*-value							
OPM	0.608	0.005	0.104	0.356	0.141	0.014	0.110
Multi-enzymes	0.536	0.144	0.856	0.345	0.011	0.043	0.340
OPM ×Multi-enzymes	0.685	0.882	0.486	0.083	0.190	0.467	0.206

^a,b^ Treatment means in a column with different superscripts differ (*p* ˂ 0.05); OPM: dietary orange peel meal; OPM × Multi-enzyme interaction effect; SEM: Standard Error of Means. Each treatment had 6 replicate pens with 4 birds each (24 birds per treatment). Pens were the experimental units for statistical analysis.

**Table 5 animals-13-02473-t005:** Effect of dietary orange peel meal with or without multi-enzymes on serum biochemistry of broiler chicks.

Factors	Glucose mg/dL	Total Protein mg/dL	Albumin mg/dL	Globulin mg/dL	Albumin/Globulin Ratio	Bilirubin mg/dL
OPM, g/kg						
0	208.00 ^ab^	5.92	1.45	4.31	0.36	0.87 ^ab^
80	185.17 ^bc^	6.93	1.21	5.72	0.23	0.71 ^b^
160	174.83 ^c^	6.67	1.52	5.17	0.46	1.09 ^a^
240	222.83 ^a^	6.72	1.62	4.95	0.37	1.03 ^a^
Multi-enzymes, g/kg						
0	180.58 ^b^	6.45	1.69	4.69	0.47	0.99
0.6	214.83 ^a^	6.66	1.20	5.38	0.24	0.85
OPM + Multi-enzymes						
0 + 0	186.00	5.47	1.18	4.29	0.30	1.01
0 + 0.6	230.00	6.37	1.73	4.33	0.41	0.72
80 + 0	175.33	7.30	1.47	5.83	0.26	0.71
80 + 0.6	195.00	6.57	0.96	5.61	0.20	0.70
160 + 0	165.30	6.80	2.06	4.77	0.74	1.24
160 + 0.6	184.33	6.73	0.96	5.57	0.19	0.95
240 + 0	195.67	6.26	2.07	3.89	0.59	1.02
240 + 0.6	250.00	7.12	1.16	6.00	0.15	1.04
SEM	12.32	0.920	0.355	1.043	0.185	0.130
*p*-value						
OPM	0.005	0.711	0.709	0.609	0.670	0.036
Multi-enzymes	0.001	0.762	0.067	0.368	0.093	0.144
OPM × Multi-enzymes	0.403	0.750	0.132	0.688	0.273	0.470

^a,b,c^ Treatment means in a column with different superscripts differ (*p* ˂ 0.05). OPM: dietary orange peel meal; OPM × Multi-enzyme interaction effect; SEM: Standard Error of Means. Each treatment had 6 replicate pens with 4 birds each (24 birds per treatment). Pens were the experimental units for statistical analysis.

**Table 6 animals-13-02473-t006:** Effect of dietary orange peel meal with or without multi-enzymes on liver enzymes and thyroid hormones of broiler chicks.

Factors	ALT U/L	AST U/L	ALT/AST	Amylase (U/L)	Protease (U/L)	T3 ng/mL	T4 ng/mL	T3/T4
OPM, g/kg								
0	22.50	19.83	1.19	64.00	53.83	1.77 ^c^	12.83	0.14
80	21.67	18.67	1.25	70.00	67.17	2.00 ^bc^	13.00	0.16
160	22.50	17.67	1.21	75.33	67.34	2.52 ^ab^	15.00	0.18
240	23.17	17.50	1.32	75.83	64.67	2.78 ^a^	14.00	0.20
Multi-enzymes, g/kg								
0	21.58	19.84	1.11	63.92	58.83	1.97 ^b^	14.08	0.14 ^b^
0.6	23.33	18.00	1.32	78.67	67.76	2.57 ^a^	13.33	0.19 ^a^
OPM + Multi-enzymes								
0 + 0	24.67	22.00	0.99	60.00	54.67	1.80	13.00	0.14
0 + 0.6	20.33	18.67	1.20	68.00	53.00	1.73	12.67	0.14
80 + 0	21.67	18.67	1.33	61.33	60.67	1.83	13.33	0.15
80 + 0.6	21.67	18.70	1.29	78.67	73.67	2.17	12.70	0.18
160 + 0	18.33	20.33	1.19	67.66	60.00	2.03	16.66	0.12
160 + 0.6	26.66	20.00	1.22	83.00	74.67	3.00	13.33	0.23
240 + 0	21.67	17.30	1.25	66.00	60.00	2.20	13.30	0.17
240 + 0.6	24.67	17.70	1.32	85.00	69.33	3.37	14.67	0.24
SEM	2.088	1.891	0.261	11.40	9.00	0.292	1.940	0.028
*p*-value								
OPM	0.913	0.061	0.052	0.706	0.414	0.011	0.664	0.220
Multi-enzymes	0.253	0.053	0.075	0.086	0.184	0.010	0.592	0.019
OPM × Multi-enzymes	0.059	0.054	0.776	0.967	0.802	0.170	0.690	0.337

^a,b,c^ Treatment means in a column with different superscripts differ (*p* ˂ 0.05). OPM: dietary orange peel meal; OPM × Multi-enzyme interaction effect; SEM: Standard Error of Means; ALT: alanine aminotransferase; AST: aspartate aminotransferase; ALT/AST: alanine aminotransferase/aspartate aminotransferase ratio; T3: triiodothyronine; T4: thyroxine; T3/T4: triiodothyronine/thyroxine ratio. Each treatment had 6 replicate pens with 4 birds each (24 birds per treatment). Pens were the experimental units for statistical analysis.

**Table 7 animals-13-02473-t007:** Effect of dietary orange peel meal with or without multi-enzymes on lipid profile of broiler chicks.

Factors	Cholesterol mg/dL	TG mg/dL	HDL mg/dL	LDL mg/dL	VLDL mg/dL
OPM, g/kg					
0	256.33 ^c^	87.83 ^a^	174.33 ^b^	64.40	17.58 ^a^
80	316.67 ^a^	58.83 ^b^	254.83 ^a^	51.06	11.77 ^b^
160	274.83 ^bc^	80.50 ^a^	208.67 ^b^	50.07	16.08 ^a^
240	298.00 ^ab^	86.33 ^a^	250.00 ^a^	34.93	17.27 ^a^
Multi-enzymes, g/kg					
0	280.50	80.67	220.25	46.70	16.14
0.6	292.42	76.08	223.67	53.53	15.21
OPM × Multi-enzymes					
0	223.67	97.00	173.33	40.86	19.43
0 + 0.6	289.00	78.67	185.33	87.93	15.73
80	330.00	61.33	263.33	56.40	17.27
80 + 0.6	303.33	66.33	246.33	45.73	15.27
160	286.33	90.00	214.33	54.00	18.00
160 + 0.6	263.33	71.00	203.00	46.13	14.17
240	282.00	74.33	240.00	35.53	14.87
240 + 0.6	314.00	98.33	260.00	34.33	19.66
SEM	13.34	4.78	16.22	11.02	0.959
*p*-value					
OPM	0.002	˂0.001	˂0.001	0.107	˂0.001
Multi-enzymes	0.224	0.194	0.769	0.393	0.188
OPM × Multi-enzymes	0.058	0.051	0.515	0.060	0.053

^a,b,c^ Treatment means in a column with different superscripts differ (*p* ˂ 0.05). OPM: dietary orange peel meal; OPM × Multi-enzyme interaction effect; SEM: Standard Error of Means; TChol: total cholesterol; TG: triglycerides; HDL: high-density lipoprotein; LDL: low-density lipoprotein; VLDL: very low-density lipoprotein. Each treatment had 6 replicate pens with 4 birds each (24 birds per treatment). Pens were the experimental units for statistical analysis.

**Table 8 animals-13-02473-t008:** Effect of dietary orange peel meal with or without multi-enzymes on antioxidant profile of broiler chicks.

Factors	GSH, mg/dL	Vit. C, mg/dL	Vit. E, µg/mL	SOD, U/mL	MDA, nmol/mL
OPM, g/kg					
0	88.83	2.24 ^b^	5.21	130.33 ^c^	73.00
80	84.67	2.61 ^ab^	5.30	139.83 ^bc^	70.50
160	103.50	2.62 ^ab^	5.51	146.67 ^ab^	65.67
240	84.50	3.37 ^a^	7.01	152.50 ^a^	68.00
Multi-enzymes, g/kg					
0	92.25	2.54	6.11	154.75 ^a^	71.67
0.6	88.50	2.88	5.42	129.92 ^b^	66.92
OPM × Multi-enzymes					
0 + 0	92.33	2.09	5.12	131.67 ^b^	77.67
0 + 0.6	85.33	2.39	5.30	129.00 ^b^	68.34
80 + 0	91.33	2.53	5.63	155.66 ^a^	70.66
80 + 0.6	78.00	2.69	4.97	124.00 ^b^	70.33
160 + 0	104.33	2.49	5.59	161.00 ^a^	65.70
160 + 0.6	102.67	2.76	5.44	132.33 ^b^	65.67
240 + 0	81.00	3.07	8.06	160.60 ^a^	72.60
240 + 0.6	88.00	3.66	5.95	134.34 ^b^	63.33
SEM	7.82	0.357	2.045	5.67	5.79
*p*-value					
OPM	0.085	0.040	0.797	0.007	0.627
Multi-enzymes	0.507	0.205	0.638	0.010	0.263
OPM × Multi-enzymes	0.621	0.941	0.945	0.038	0.743

^a,b,c^ Treatment means in a column with different superscripts differ (*p* ˂ 0.05). OPM: dietary orange peel meal; OPM × Multi-enzyme interaction effect; SEM: Standard Error of Means; GSH: glutathione; Vit. C: ascorbic acid; Vit. E: tocopherol; SOD: superoxide dismutase; MDA: malondialdehyde. Each treatment had 6 replicate pens with 4 birds each (24 birds per treatment). Pens were the experimental units for statistical analysis.

## Data Availability

The datasets used and/or analyzed during the current study are available from the corresponding author on reasonable request.

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
