# Peer review of "Effect of Dietary Orange Peel Meal and Multi-Enzymes on Productive, Physiological and Nutritional Responses of Broiler Chickens"

_animals, 2023, doi:10.3390/ani13152473_

Round 1

Reviewer 1 Report (Previous Reviewer 1)

Dear authors,

thank you for your second review of this manuscript.

Best regards

Author Response

Thank you very much for your positive comments.

Reviewer 2 Report (Previous Reviewer 2)

The authors made a significant revision to the manuscript. It can be considered for publication if authors address my comments as follows:

- In Table 1, authors mentioned only 2 phases: stater and grower, but in simple summary and abstract (L22-24, L40), a finisher period is mentioned. 

- For determination of nutrient digestibility, authors must provide more detail of the method. What method did you used to measure the digestibility, using marker or total collection? And what did you treat the collected samples? Need to detail

- Authors should group serum enzyme activities in a separate table for easy following. Currently, table 6 shows liver enzymes and metabolites (thyroid hormones), then table 8 again illustrates amylase and protease.

- In table 2, it is not body weight gain, it must be body weight (accumulated body weight, or body weight at that measurement)

- When the significance of Multi-enzyme is observed, it does need to use superscript to indicate because you have only 2 levels then it is obvious can see the difference from the table. BUT, when both significances of OPM and interaction are detected, you need to use different superscripts (abc vs ABC), otherwise, it could cause confusion (only table 2)

- Keep consistent when presenting the unit of parameters. For example, Table 1, ME (Kcal/kg). But CP%, then Table 2, Feed intake, g

Author Response

Response to Reviewer 2 Comments

Point 1: The authors made a significant revision to the manuscript. It can be considered for publication if authors address my comments as follows:

Response 1: Thank you very much for your comments; we have addressed all the suggestions in track changes in the revised manuscript.

Point 2: - In Table 1, authors mentioned only 2 phases: stater and grower, but in simple summary and abstract (L22-24, L40), a finisher period is mentioned. 

Response 2: Thanks for your comment, corrected as requested.  There were only 2 phases, starter and grower phase.

Point 3: - For determination of nutrient digestibility, authors must provide more detail of the method. What method did you used to measure the digestibility, using marker or total collection? And what did you treat the collected samples? Need to detail

Response 3: Total excreta collection method was used. More details of the methods about the nutrient digestibility added as requested in L163-179.

Point 4: Authors should group serum enzyme activities in a separate table for easy following. Currently, table 6 shows liver enzymes and metabolites (thyroid hormones), then table 8 again illustrates amylase and protease.

Response 4: Thank you for your suggestion. The serum enzymes added in the similar table as suggested along with liver enzymes, please see Table 6.

Point 5: - In table 2, it is not body weight gain, it must be body weight (accumulated body weight, or body weight at that measurement)

Response 5: Thank you very much for your comments. There was mistyping in few values in table 2 that we have corrected now. These corrected values are highlighted in red color in the table 2. The data provided in table 2 are related to BWG during different periods.

Point 6: - When the significance of Multi-enzyme is observed, it does need to use superscript to indicate because you have only 2 levels then it is obvious can see the difference from the table. BUT, when both significances of OPM and interaction are detected, you need to use different superscripts (abc vs ABC), otherwise, it could cause confusion (only table 2)

Response 6:  Thank you for your suggestions and inputs. The main effects of Multi-enzyme are significant in the revised table 2, therefore superscript was used. In Table 2, when there was an effect of OPM levels, as for FCR for 22-42 d, we used superscripts (a, b, c) to compare the 4 levels of OPM that was mentioned in first part of the Table. No interaction effect of OPM and multienzyme was observed in Table 2, and so we didn’t use superscripts (a, b, c). We don’t think it could cause any confusion, and this was one of the reasons to separate and mention in Tables the OPM effects, multienzyme effects, and interaction effects, separately.

 Point 7: Keep consistent when presenting the unit of parameters. For example, Table 1, ME (Kcal/kg). But CP%, then Table 2, Feed intake, g

Response 7: These are the standard units in dietary formulations. In Table 1, ME (Kcal/kg) this is the unit of metabolizable energy. The CP is in percentage and other analyzed nutrients are expressed in percentage as well. We chose not to present in g/kg as it will increase the Table size and will create formatting issues. The performance data including BWG, and feed intake are presented in grams, and that is how it is presented in Tables. We understand it may appear inconsistent, but it is better understandable. But thank you for your comments.

This manuscript is a resubmission of an earlier submission. The following is a list of the peer review reports and author responses from that submission.

Round 1

Reviewer 1 Report

Dear authors,

you presented a research related to the effect of orange peel mixed in broiler feed, with or without enzymes to the production, physiology and immune response of the broilers.

General comments:

-title should be changed (with or without multiple-enyzmes should be changed; Latin name of orange should be added)

- please pay attention to the first capital letter for some enzymes, biochemical parameters etc.

- did the broilers received OPM in the feed all through the experiment, both in starter and grower diets? what about the enzymes?

-please uniform in the whole text enzyme or multi-enzyme

-if possible, mention the mortality ratio during the experiment

Specific comments:

-L16-17 sentence should be rephrased ("source of improving profilability"?)

- L 23 (and in the whole text) - level of OPM; use another word instead of level

- L27-30 those sentences needs to be rephrased

-L 31 (and in the whole text) - 6 weeks of age (the duration of the experiment was 6 weeks or until the birds were 6 weeks old)? L 87- 42 days of age?

- L 35- broiler diets supplemented with OPM had considerable lower serum level

- L 49 secondary nutrients - please elaborate

- L78-81 this is a repeat of the Introduction

-L 81-89 needs to be rephrased or maybe combined with other chapters in the Material and Methods

- L 101 - orange peel meal was acquired from local vendors or orange peel?

-L 106 and 85 and 110- please unify the name of the enzyme mixture (Optiphos or Opti-phos)

-L110 - gm is abbreviation for?

-L 145 - by the weight of the live weight? needs to be rephrased

-L 169 twice using

-L176 twice supplementation? please rephrase

Table 1. two control groups are mentioned here as OPM 0 and Enzyme 0? an additional control group is the one mentioned as OPMxEnzymes as 0? Also,under Enzyme is written 0,5 - is this the same group as under OPMxEnzymes 0+0,5? you had altogether 14 groups in your experiment (what does the factor means in this table?) (the same for Table 3, 4, 5, 6, 7, 8)

-L 226-serum biochemistry? please rephrase

-L260-261 please write reference for this statement

-L 274 "across dietary dry orange peel powder"? please rephrase

-L 293 - feed feeding value? please rephrase

-L 301 different treatments supplemented? please rephrase; also, write in italic the Latin name

-L 305-308 paragraph need revision, dot missing in between the sentences

-L 317-318 please rephrase (it was discovered?)

-L 325 oilve?

-L 332 broilers fed OMP levels?

-L 331-331 is the same as L 317-318

Author Response

Please see attached file for our response to your comments. Thank you. 

Response to Reviewer 1 Comments

Dear authors,

you presented a research related to the effect of orange peel mixed in broiler feed, with or without enzymes to the production, physiology and immune response of the broilers.

Dear Respective Reviewer, thank you very much for your comments and inputs which has improved the final form of our manuscript. Please find our response to your comments below and a list of changes that we made according to your suggestions. We have addressed all the comments in the revised manuscript.   Authors’ responses are in red color.

General comments:

Point 1: -title should be changed (with or without multiple-enyzmes should be changed; Latin name of orange should be added)

Response 1: Done as requested; Latin name of orange is not required as it will confuse readers.

Point 2: - please pay attention to the first capital letter for some enzymes, biochemical parameters etc.

Response 2: Done as requested

Point 3: - did the broilers received OPM in the feed all through the experiment, both in starter and grower diets? what about the enzymes?

Response 3: The OPM with or without enzymes were added during the starter and grower periods

Point 4: -please uniform in the whole text enzyme or multi-enzyme

Response 4: Done as suggested throughout the manuscript.

Point 5: -if possible, mention the mortality ratio during the experiment

Response 5: No mortality was observed

Specific comments:

Point 6: -L16-17 sentence should be rephrased ("source of improving profilability"?)

Response 6: Done as requested, please see Lines 16-17

Point 7: - L 23 (and in the whole text) - level of OPM; use another word instead of level

Response 7: Done as suggested

Point 8: - L27-30 those sentences needs to be rephrased

Response 8: Done as requested, please see L28-32

Point 9: -L 31 (and in the whole text) - 6 weeks of age (the duration of the experiment was 6 weeks or until the birds were 6 weeks old)? L 87- 42 days of age?

Response 9: changed to days

Point 10: - L 35- broiler diets supplemented with OPM had considerable lower serum level

Response 10: Done as requested L 48

Point 11: - L 49 secondary nutrients - please elaborate

Response 11:  changed L 63

Point 12: - L78-81 this is a repeat of the Introduction

Response 12:  Deleted as you suggested.

Point 13: -L 81-89 needs to be rephrased or maybe combined with other chapters in the Material and Methods

Response 13:  deleted as requested

Point 14: - L 101 - orange peel meal was acquired from local vendors or orange peel?

Response 14: Oranges were acquired from local vendors L123.

Point 15: -L 106 and 85 and 110- please unify the name of the enzyme mixture (Optiphos or Opti-phos)

Response 15: Done as suggested L 105-108.

Point 16: -L110 - gm is abbreviation for?

Response 16: for gram

Point 17: -L 145 - by the weight of the live weight? needs to be rephrased

Response 17: Done as requested L 183-184.

Point 18: -L 169 twice using

Response 18: Done as requested L186

Point 19: -L176 twice supplementation? please rephrase

Response 19: Rephrased as requested L208.

Point 20: Table 1. two control groups are mentioned here as OPM 0 and Enzyme 0? an additional control group is the one mentioned as OPMxEnzymes as 0? Also,under Enzyme is written 0,5 - is this the same group as under OPMxEnzymes 0+0,5? you had altogether 14 groups in your experiment (what does the factor means in this table?) (the same for Table 3, 4, 5, 6, 7, 8)

Response 20: Thank you for your comment, For Table 1, there was only one control for the treatments during the starter or grower periods. Four dietary orange peel meal (OPM) levels were supplemented, namely control (without OPM) and with 80, 160, and 240 grams/kg of diet. Each of these diets was repeated with added multi-enzymes. To clarify, we mentioned clearly in the text (Lines 113-115), the eight treatment diets used in this study.

Point 21: -L 226-serum biochemistry? please rephrase

Response 21: changed L282

Point 22: -L260-261 please write reference for this statement

Response 22: Done L 357, L358

Point 23: -L 274 "across dietary dry orange peel powder"? please rephrase

Response 23: rephrased as requested L357-361

Point 24: -L 293 - feed feeding value? please rephrase

Response 24: Done L392

Point 25: -L 301 different treatments supplemented? please rephrase; also, write in italic the Latin name

Response 25: the statement was rephrased L397-402. 

Point 26: -L 305-308 paragraph need revision, dot missing in between the sentences

Response 26: Rephrased L406-408

Point 27: -L 317-318 please rephrase (it was discovered?)

Response 27: Done L421

Point 28: -L 325 oilve?

Response 28: Corrected 430

Point 29: -L 332 broilers fed OMP levels?

Response 29: Corrected L454

Point 30: -L 331-331 is the same as L 317-318

 Response 30: Done L457-458

Reviewer 2 Report

This manuscript can be considered for publication in Animals if a major revision is made. Please find my comments below for your correction.

- The introduction did not tell the reason for enzyme supplementation, the type of enzyme used in the experiment

- L60-62: '...without negative impact...', so why did authors investigate the use of enzyme

- L78-80: this replaced the aims. It did not belong to this section.

- L101-105: authors mentioned OPM was proximate analysis but in L115-116, authors cited results from previous publication?

- Type and similar of orange

- In all table, superscripts don't need to apply for significant effect of enzyme because there are two level, with or without. However, superscripts must be applied where interaction was found significant.

- L35 and forward: don't use the statement "Broiler diets supplemented with OPM with or without enzymes had considerably". Just mention OPM is enough because you are talking about the main of OPM, not enzyme or interaction. 

The value of LDL (P=0.107) in table 7 is not supported the conclusion in the abstract

- L37-38: in here, authors talked about interaction but the results from Table 6 did not support the statement. In fact, it was the single effect of the enzyme

- Table 2 and 8: for FCR during 22-42 and for SOD activity, superscripts need to be added in the interaction results (significant interactions)

- L40-44: the conclusion of the abstract did not based on the result

OPM and enzyme combination (interaction) did not effect any parameters, except (FCR during 22-42, not the overall period) and SOD activity

- Similarly the abstract, the conclusion section did not follow the results

The authors did not conclude what levels of OPM or enzyme should be used. In case supplement both, at which levels are beneficial

Author Response

Response to Reviewer 2 Comments

This manuscript can be considered for publication in Animals if a major revision is made. Please find my comments below for your correction.

Dear Respective Reviewer, thank you very much for your comments and inputs which has improved the final form of our manuscript. Please find our response to your comments below and a list of changes that we made according to your suggestions. We have addressed all the comments in the revised manuscript.   Authors’ responses are in red color.

Point 1:- The introduction did not tell the reason for enzyme supplementation, the type of enzyme used in the experiment

Response 1:  available information added at L87-95

Point 2: - L60-62: '...without negative impact...', so why did authors investigate the use of enzyme

Response 2:  this means without any side effects. Changed L76

Point 3: - L78-80: this replaced the aims. It did not belong to this section.

Response 3: deleted L102-104

Point 4: - L101-105: authors mentioned OPM was proximate analysis but in L115-116, authors cited results from previous publication?

Response 4: Only ME was calculated according to Pauzenga [10]. The sentence is now divided to make it clear L 152-153

Point 5: - Type and similar of orange

Response 5:  the type added 

Point 6: - In all table, superscripts don't need to apply for significant effect of enzyme because there are two level, with or without. However, superscripts must be applied where interaction was found significant.

Response 6:  thanks for your comments, done as suggested. Please see all tables

Point 7: - L35 and forward: don't use the statement "Broiler diets supplemented with OPM with or without enzymes had considerably". Just mention OPM is enough because you are talking about the main of OPM, not enzyme or interaction. 

Response 7: Done as suggested throughout the manuscript.

Point 8: The value of LDL (P=0.107) in table 7 is not supported the conclusion in the abstract

Response 8: thank you for your comments, therefore we modified our statement. Please see lines 47-50 in abstract.

Point 9: - L37-38: in here, authors talked about interaction but the results from Table 6 did not support the statement. In fact, it was the single effect of the enzyme

Response 9: corrected as suggested

Point 10:- Table 2 and 8: for FCR during 22-42 and for SOD activity, superscripts need to be added in the interaction results (significant interactions)

Response 10: thank you; superscripts added as suggested.

Point 11: - L40-44: the conclusion of the abstract did not based on the result

Response 11: The conclusions are changed as requested.

Point 12: OPM and enzyme combination (interaction) did not effect any parameters, except (FCR during 22-42, not the overall period) and SOD activity

Response 12: Yes, we do agree, statements changed accordingly.

Point 13:- Similarly the abstract, the conclusion section did not follow the results

Response 13: Statements changed as requested, please see conclusion

Point 14: The authors did not conclude what levels of OPM or enzyme should be used. In case supplement both, at which levels are beneficial

Response 14:  Thank you so much for your suggestion. We changed the conclusions and considered your suggestions. 

Reviewer 3 Report

The paper is very well written, and characterizes the effect of supplemented different doses of orange peel powder and enzymes on the productive performance of the chickens. This is a meaningful study . However, there are some problems, which must be solved before it is considered for publication.

line81:change"240 day-old Arbor Acres broiler chicks" to "240 one day-old Arbor Acres broiler chicks;

line94: the pen used in the trail  is with dimensions of 130, 80, and 70 cm in length, width, and height.But there would be a problem for only 5 bird raised per pen, especially in starter period.

Line105,it would be better to present the the component of OPM in the manuscript.

Table 1, the group name showed in table 1 were 1,2,3,4. But the group name in the results part is different. And in line 84, two levels of enzymes inclusion (0 or combination of 0.5 g of Nutrikem and 0.1 g Optiphos per kg diet) were added. What the 0.6% of enzyme inclusion were substitute? In Table 2,3,4,5,6,7,8,it all display  0.5 g/kg as the enzyme dose, so it is 0.5 or 0.6?

Line 131, one birds from each repelicate were housed alone in the metabolic cage or housed together? this should clarified. If six bird housed together, only one data from each treatment is not sufficient to subject statistic.

Line 136: 08:00 am?

line 141:24 birds from each treatment means take four chickens form each replicates or 24 birds were randomly selected from the group?

Line 171: P value should be add after every significant  results at the end of sentence , such as line179(significantly improve), line 181(did not impact). Please check the whole result part.

Table 2: why the author prefer to present Body weight gain and feed intake, not average daily weight gain and average daily feed intake. Althoug it is actullay almost the same. The mortality was measured in material and methods but not present the results.

line 246: the abbreviation of  ascorbic acid and superoxide dismutase were displayed in the material and methods part.

Table 8, "<.0.001" should be a writing error.

Author Response

Response to Reviewer 3 Comments

The paper is very well written, and characterizes the effect of supplemented different doses of orange peel powder and enzymes on the productive performance of the chickens. This is a meaningful study . However, there are some problems, which must be solved before it is considered for publication.

Greetings, reviewer, we very much appreciate your comments, which has helped to strengthen and enhance the manuscript's final form. Please see the modifications as track changes in the manuscript and our response to your comments below. We have addressed all the comments/suggestions in the revised manuscript. Authors’ responses are in red color.

Point 1: line81:change"240 day-old Arbor Acres broiler chicks" to "240 one day-old Arbor Acres broiler chicks;

Response 1: Done as suggested on L 34-35

Point 2: line94: the pen used in the trail  is with dimensions of 130, 80, and 70 cm in length, width, and height.But there would be a problem for only 5 bird raised per pen, especially in starter period.

Response 2: We appreciate your comments, but during the trail periods, we had no problems with the birds.

Point 3: Line105,it would be better to present the the component of OPM in the manuscript.

Response 3: the chemical analysis is added as footnotes in Table 1.

Point 4: Table 1, the group name showed in table 1 were 1,2,3,4. But the group name in the results part is different. And in line 84, two levels of enzymes inclusion (0 or combination of 0.5 g of Nutrikem and 0.1 g Optiphos per kg diet) were added. What the 0.6% of enzyme inclusion were substitute? In Table 2,3,4,5,6,7,8,it all display  0.5 g/kg as the enzyme dose, so it is 0.5 or 0.6?

Response 4: Thank you for your comments. The group names in table 1 are changed now and enzyme level added in L 40 and L 112

Point 5: Line 131, one birds from each repelicate were housed alone in the metabolic cage or housed together? this should clarified. If six bird housed together, only one data from each treatment is not sufficient to subject statistic.

Response 5: one per replicate that represented the pen in each metabolic cage alone, L169-170

Point 6: Line 136: 08:00 am?

Response 6: yes am, L174

Point 7: line 141:“24 birds from each treatment” means take four chickens form each replicates or 24 birds were randomly selected from the group?

Response 7: 24 birds were selected from each treatment group (4 birds each replicate), L180

Point 8: Line 171: P value should be add after every significant  results at the end of sentence , such as line179(significantly improve), line 181(did not impact). Please check the whole result part.

Response 8: thank you, done as suggested.

Point 9: Table 2: why the author prefer to present Body weight gain and feed intake, not average daily weight gain and average daily feed intake. Althoug it is actullay almost the same. The mortality was measured in material and methods but not present the results.

Response 9: We prefer to use body weight gain and feed intake. This way it is easy for readers to understand. We did not observe any deaths during the trail.

Point 10: line 246: the abbreviation of  ascorbic acid and superoxide dismutase were displayed in the material and methods part.

Response 10: Done L 197-198.

Point 11: Table 8, "<.0.001" should be a writing error.

Response 11: correct, thank you so much for your suggestions and inputs which has helped to improve the final form of our manuscript.

Round 2

Reviewer 1 Report

Dear authors,

thank you for taking into consideration all the comments given by the reviewer. However, there are still some changes needed:

- L 96-97 you can combine these two sentences in one

- L 122 Nutrikem dry is the same product as  Nutrikem XL Pro?

- L126 abbreviation for gram is g and not gm

- L 381 jejunum and ileum were increased? maybe to use another word instead of increased (or explain what does increased mean exactly)

- L385 mulit?

Author Response

Response to Reviewer 1 Comments (Round 2)

Point 1: thank you for taking into consideration all the comments given by the reviewer. However, there are still some changes needed:

Response 1: Thank you very much for your comments; we have addressed all the comments and suggestions as track changes in the revised manuscript.

Point 2: L 96-97 you can combine these two sentences in one

Response 2: Done as requested, please see L106-107.

Point 3: L 122 Nutrikem dry is the same product as  Nutrikem XL Pro?

Response 3: Yes, but now we deleted “dry” to remove the confusion. Please see Lines 132-133.

Point 4: - L126 abbreviation for gram is g and not gm

Response 4: Done L136

Point 5: - L 381 jejunum and ileum were increased? maybe to use another word instead of increased (or explain what does increased mean exactly)

Response 5: Corrected as requested, L395-396

Point 6: - L385 mulit?

Response 6:  typed incorrectly, now corrected as requested, Multi – L399.

Reviewer 2 Report

Thanks for addressing my comments. However, authors must pay more attention to the result interpretation. The interaction between OPM and enzyme didn't show any significance on digestibility or abdominal fat (only single effects were observed). It never can be concluded as the L46-49 and L426-429 based on the observed results.

Also I didn't see any where authors measured and presented urea and creatinine concentration, but in L391 author talked about them, and in the conclusion as well.

Author Response

Response to Reviewer 2 Comments (Round 2)

Point 1: Thanks for addressing my comments. However, authors must pay more attention to the result interpretation. The interaction between OPM and enzyme didn't show any significance on digestibility or abdominal fat (only single effects were observed). It never can be concluded as the L46-49 and L426-429 based on the observed results.

Response 1: thank you very much for your comments; we have addressed all suggestions related to the results in the revised manuscript. Please see L45-49, L254-256, L 273-274, and L 435-444

Point 2: Also I didn't see any where authors measured and presented urea and creatinine concentration, but in L391 author talked about them, and in the conclusion as well.

Response 2: Corrected as requested, L400, L 407.
